# A Study on the Relationship between the Design of Aerotrain and Its Stability Based on a Three-Dimensional Dynamic Model †

**Quang Huan Luong** [1,*] **, Jeremy Jong** [2] **, Yusuke Sugahara** [1] **and Daisuke Matsuura** [1] **and Yukio Takeda** [1]

1   School of Engineering, Tokyo Institute of Technology, Tokyo 152-8552, Japan;
    sugahara.y.ab@m.titech.ac.jp (Y.S.); matsuura.d.aa@m.titech.ac.jp (D.M.); takeda.y.aa@m.titech.ac.jp (Y.T.)
2   Graduate School of Engineering, Kokushikan University, Tokyo 154-8515, Japan; jeremyjong@gmail.com
*   Correspondence: luong.h.aa@m.titech.ac.jp; Tel.: +81-80-9534-3985
†   This paper is an extended version of our paper published in Luong, Q.H.; Jong, J.; Sugahara, Y.;
    Matsuura, D.; Takeda, Y. A 3-Dimensional Dynamic Model of the Aerotrain and the Horizontal Tail Effect on
    the Longitudinal Stability. In Proceedings of the IFToMM International Symposium on Robotics and
    Mechatronics, Taipei, Taiwan, 28–30 October 2019.

**Abstract:** A new generation electric high-speed train called Aerotrain has levitation wings and levitates under Wing-in-Ground (WIG) effect along a U-shaped guideway. The previous study found that lacking knowledge of the design makes the prototype unable to regain stability when losing control. In this paper, the nonlinear three-dimensional dynamic model of the Aerotrain based on the rigid body model has been developed to investigate the relationship between the vehicle body design and its stability. Based on the dynamic model, this paper considered an Aerotrain with a horizontal tail and a vertical tail. To evaluate the stability, the location and area of these tails were parameterized. The effects of these parameters on the longitudinal and directional stability have been investigated to show that: the horizontal tail gives its best performance if the tail area is a function of the tail location; the larger vertical tail area and (or) the farther vertical tail location will give better directional stability. As for the lateral stability, a dihedral front levitation wing design was investigated. This design did not show its effectiveness, therefore a control system is needed. The obtained results are useful for the optimization studies on Aerotrain design as well as developing experimental prototypes.

**Keywords:** wing-in-ground effect; aerotrain; aerial robotics

## 1. Introduction

In recent years, the expansion of the range, transportation mass, types of activities undertaken by human beings has created global environmental problems. Therefore, a transportation system with high-efficiency and low power consumption is one of the important solutions for these problems and will benefit society. For this purpose, in the future, the next-generation transportation system must include two significant features, namely, high speed and high efficiency [1]. One of the potential candidates is the MAGLEV train which could reach maximum velocity of up to 603 km/h [2]. However, in spite of the advantages of the technology, the MAGLEV train system has smaller economic efficiency value than that of the Aerotrain [3], a Wing-in-Ground effect vehicle, proposed by Kohama et al. [4] at Tohoku University, Japan.

The Aerotrain, as shown in Figure 1, has wings and levitates on a concrete U-shaped guideway by the Wing-In-Ground (WIG) effect. The advantage of a low-cost guideway is expected to reduce the maintenance cost of the track that up to 30% of the total operation costs according to a High-speed rail

feasibility study reported by Rocky Mountain Rail Authority [5]. Compared with normal MAGLEV high-speed trains, the massive energy required to maintain the electrical magnetic force is not needed, which means that the Aerotrain could reduce a lot of power consumption. The difference between an airplane and the Aerotrain is the WIG effect. The WIG effect happens when an airfoil or wing-like object is close to the ground as shown in Figure 2 when the altitude h is less than 25% of the wing chord c [6], a high-pressure air cushion will be created for the stagnation of the air which generates a greater lift force and a greater lift-drag ratio (L/D) than the airfoil at higher altitude. When travelling, the pitching and rolling motion of the vehicle are controlled through the control surfaces on the levitation wings. The aerodynamic lift generated from levitation wings will levitate the train very close to the ground under WIG effect. Thus, a greater lift force under WIG effect benefits the Aerotrain a higher aerodynamic efficiency than that of conventional airplanes. This same principle is adopted to the guide wing for keeping the position of the train at the center of the guideway and keeping the direction parallel to the guideway. By changing the angle of the control surface on the guide wing (the vertical wings at the tip of the levitation wings), the side force and the yawing moment are also changed to make the Aerotrain stay on the proper track. The WIG effect could also provide a recovering moment to ensure the stability of the Aerotrain when an external disturbance in the rolling axis of motion happens.

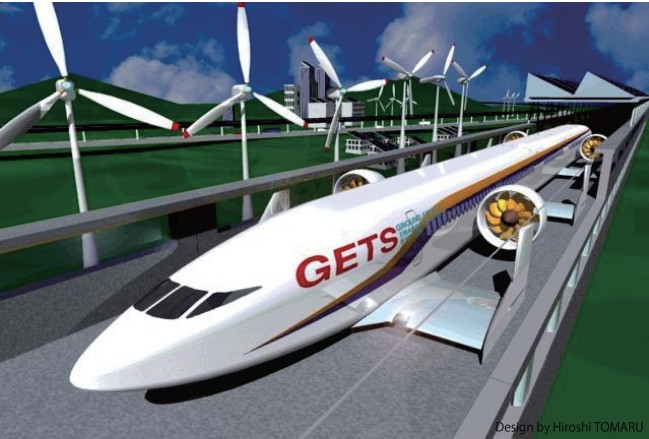

**Figure 1.** Aerotrain concept [7].

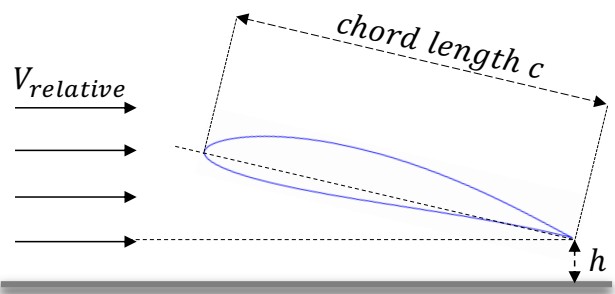

**Figure 2.** Wing in ground (WIG) effect.

The development of WIG effect vehicles in general and WIG crafts (operating on water surface) in particular have showed rapidly technical improvement in recent years. Hameed [8] and Matdaud [9] have an extensive literature review on the design and the existing stability control system technology's development and enhancement for WIG crafts. According to the reviews, each WIG craft configuration shows its own main design features, advantages and yet disadvantages. With the presence of the disturbance caused by the round wave condition, stability problems arise that make safety issues of operating WIG crafts become more delicate to handle. Compared with those conventional WIG

crafts [10–13], the Aerotrain has less external disturbances and it is safer because the operation of the vehicle is based on the use of a solid guideway [3]. There have been several studies on the control of the Aerotrain [14–16], but there is insufficient knowledge on the design of the vehicle body which has a desirable dynamic characteristic. Thus, the design of the Aerotrain with only levitation wings and guide wings is controllable, but does not have dynamic stability. A recent study [3] found the influence of the relative positions of the levitation wings and the angle of attack on the aerodynamic performances of the Aerotrain. However, in an experiment of the previous research, the prototype could not regain stability as it was pitching the nose up until it became stall when losing its control [17]. That potential risk raised a question about the relationship between the body design of the vehicle and its stability. Therefore, to answer the question, the long-term objective of this research is to investigate the relationship between the Aerotrain's design parameters and its stability.

Generally, the longitudinal stability plays an important role in the stability of the vehicle and it could be studied separately with directional and lateral stability. Moreover, as for a flying vehicle which have a symmetrical shape about the centerline, moderate changes in angle of attack will have little or no influence upon the yaw or roll. This permits the stability and control analysis to be divided into longitudinal and lateral-directional analysis [18]. According to this assumption, the previous study [19] found that the horizontal tail has a great contribution on the longitudinal stability of the Aerotrain by introducing a two-dimensional dynamic model. The purpose of this paper is to deeply clarify the design-stability relation in a complicated condition when longitudinal, lateral and directional stability are all considered. In this paper, firstly, a three-dimensional nonlinear dynamic model of the Aerotrain with a horizontal tail and a vertical tail has been introduced. Secondly, based on the dynamic model, the effect of the horizontal tail design (horizontal tail location and horizontal tail area), the vertical tail design (vertical tail location and vertical tail area) and levitation wing design (flat levitation wing and dihedral levitation wing) to the dynamic stability of the Aerotrain has been evaluated through several indices that will be explained later in Section 3.

## 2. Dynamic Model of the Aerotrain

Considering it as a flight vehicle, the attitude of the Aerotrain is described in a body-fixed coordinate system $O - XYZ$ in which the origin is the center of gravity of the vehicle. The $X$-axis is in the plane of symmetry of the body and points forward. The $Z$-axis is in the same plane as $X$-axis, perpendicular to $X$-axis and points down. The $Y$-axis is perpendicular to the symmetry plane and points out the right wing. However, only using the body-fixed coordinate system is not enough to express the gravitational force acting on the vehicle. Thus, the orientation of the body-fixed coordinate system with respect to the gravity vector needs to be determined as shown in Figure 3. This orientation can be specified using the Euler angle (pitch angle $\theta$, roll angle $\phi$ and yaw angle $\psi$) of the body-fixed coordinate system with respect to an inertial system $O_E - X_E Y_E Z_E$, where the inertial system is oriented such that the $Z_E$-axis points down (parallel to the gravity vector), the $X_E$-axis points North and the $Y_E$-axis completes the right-handed system, and therefore points East.

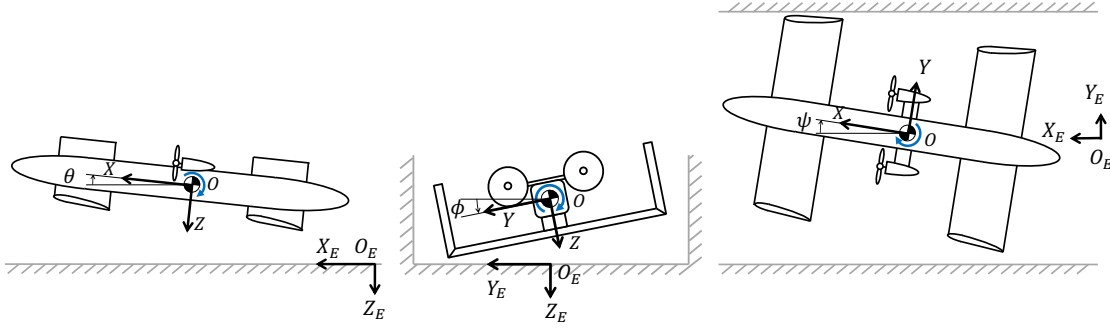

**Figure 3.** Coordinate systems and Euler angles.

The Aerotrain is considered as a rigid body in steady state with a total of 6 degrees of freedom which include 3 translations and 3 rotation motions. The 3 translations are caused by the aerodynamic forces and the gravity force. These forces and aerodynamic moments are the reason of the 3 rotational motions which are pitching, rolling and yawing moment. In the Aerotrain model, the aerodynamic lifts generated from two levitation wings and 4 guide wings are under the WIG effect during cruise speed. Given that the operation occurs inside a U-shaped guideway, the movement in Z-axis and Y-axis and the rotation about 3 axes are more important than the movement in X-axis. Thus, in this paper, the 1 degree of freedom of the X-axis translation has been ignored under the assumption that the vehicle travels at constant speed. Therefore, Z-axis and Y-axis forces as well as 3 moments (about 3 axes) have been modelled.

### 2.1. Wing-in-Ground Effect Model

According to Honda [20], in the area affected by the ground effect, the lift coefficient changes nonlinearly, but this can be represented by the sum of the nonlinear term caused by the influence of the ground effect and the linear term unrelated to the influence of the ground effect.

$$L = \frac{1}{2}\rho V_{x0}^2 S C_L \tag{1}$$

$$C_L = C_{Lg.e} + C_{Lnormal}, \tag{2}$$

where $L$ is the aerodynamic lift, $C_L$ is the lift coefficient, $\rho$ is the air density, $V_{x0}$ is the velocity at steady state and $S$ is the wing area. The lift coefficient under the WIG effect $C_{Lg.e}$ and in normal condition $C_{Lnormal}$ could be expressed as followings :

$$C_{Lg.e} = a_1 \exp\left(-a_2 \frac{h}{c}\right) \tag{3}$$

$$C_{Lnormal} = a_3 \alpha + a_4, \tag{4}$$

where $a_1, a_2, a_3, a_4$ are obtained constants from the curve fitting of the experimental results, and they are different at the front and rear levitation wings as can be seen in Figure 4; $\alpha$ is the angle of attack. In Figure 4, $C_{Lf}, C_{Lr}$ are the aerodynamic lift coefficient of the front and rear levitation wings; $h_f, h_r$ are the height of the front and rear levitation wings; $\alpha_f, \alpha_r$ are the angle of attack of the front and rear levitation wings.

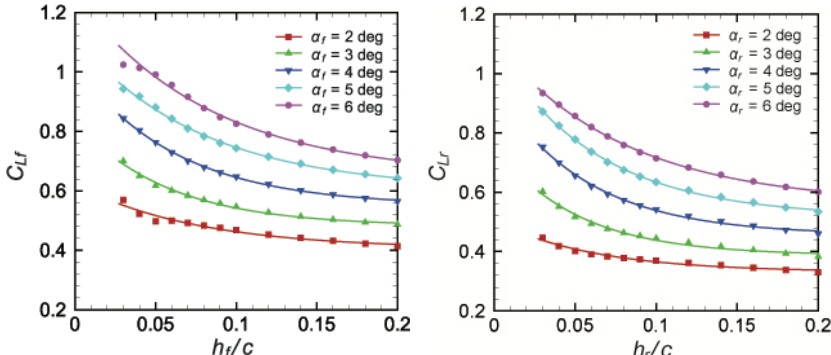

**Figure 4.** Aerodynamic lift coefficient of the front and rear levitation wings under the WIG effect [20].

Similar to the aerodynamic lift, the aerodynamic moment under WIG effect is nonlinear, and it can be represented by the sum of the nonlinear term caused by the WIG effect and the linear term unrelated to the influence of the WIG effect.

$$M = \frac{1}{2}\rho V_{x0}^2 S c C_M \tag{5}$$

$$C_M = C_{Mg.e} + C_{Mnormal}, \tag{6}$$

where $m$ is the aerodynamic moment, $C_M$ is the moment coefficient. The moment coefficient under WIG effect $M_{Mg.e}$ and in normal condition $M_{Mnormal}$ could be expressed as followings:

$$C_{Mg.e} = a_5 \exp\left(-a_6 \frac{h}{c}\right) + a_7 \exp\left(-a_8 \frac{h}{c}\right) \tag{7}$$

$$C_{Mnormal} = a_9 \alpha + a_{10}, \tag{8}$$

where $a_5, a_6, a_7, a_8, a_9, a_{10}$ are obtained constants from the curve fitting of the experimental results as can be seen in Figure 5, and they are also different at the front and rear levitation wings [20]. $C_{Mf}$ in the Figure 5 is the aerodynamic coefficient of the front levitation wing.

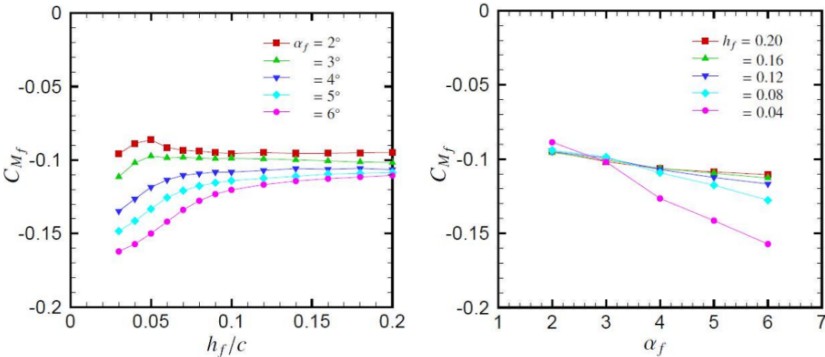

**Figure 5.** Aerodynamic moment coefficient of the front levitation wing under the WIG effect [20].

### 2.2. Aerodynamic Force along the Z-Axis

The aerodynamic force along the Z-axis includes the lift force from the front and the rear levitation wing of the Aerotrain. Figure 6 shows the differential slice $dy_i$ with its height $h_f$ at the spanwise $y_i$ of the front levitation wing with span $b$ at the roll angle $\phi$ that will be later used to calculate the differential lift. $h_0$ is the height of the Center of Gravity (CoG) of Aerotrain at the steady state, and $h$ is the displacement of the CoG from the steady state in the Z-axis.

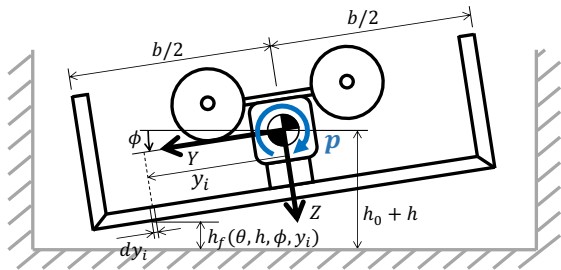

**Figure 6.** The front view of Aerotrain [21].

Under the influence of the WIG effect, the lift coefficient of the front wing $C_{Lfi}$ and the rear wing $C_{Lri}$ at the spanwise $y_i$ could be written as:

$$C_{Lfi}(y_i) = C_{Lg.e} + C_{Lnormal} = a_1 \exp\left(-a_2 \frac{h_f}{c_f}\right) + a_3 \alpha_f + a_4 \tag{9}$$

$$C_{Lri}(y_i) = C_{Lg.e} + C_{Lnormal} = b_1 \exp\left(-b_2 \frac{h_r}{c_r}\right) + b_3 \alpha_r + b_4. \tag{10}$$

The parameters $a_1, ..., a_4, b_1, ..., b_4$ are defined by the curve fitting from the experiment [20]; $c_f, c_r$ are the chord length of the front and rear levitation wing relatively. Assume that the pitch

angle $\theta$ and roll angle $\phi$ are small enough, from the geometrical relationship shown in Figure 7, the height of the front wing $h_f$ and the rear wing $h_r$ are the linearized function of the actual height $h$ of the CoG, the pitch angle $\theta$, the roll angle $\phi$ and the location $y_i$:

$$h_f(h, \theta, \phi, y_i) = h_0 + h + l_f\theta - y_i\phi \tag{11}$$

$$h_r(h, \theta, \phi, y_i) = h_0 + h - l_r\theta - y_i\phi, \tag{12}$$

where $l_f, l_r$ are the distance from the CoG to the aerodynamic center which is located at the quarter-chord from the leading edge of the front and the rear levitation wing as shown in Figure 7.

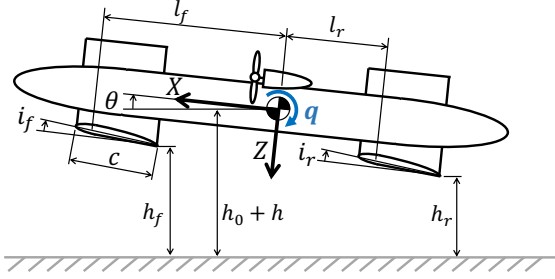

**Figure 7.** The side view of Aerotrain [21].

The Angle of Attack (AOA) of the front wing $\alpha_f$ and the rear wing $\alpha_r$ are the function of the AOA $\alpha$ and the pitching angular velocity $q$:

$$\alpha_f(\alpha, q) = \alpha + i_f - q\frac{l_f}{V_{x0}} \tag{13}$$

$$\alpha_r(\alpha, q) = \alpha + i_r + q\frac{l_r}{V_{x0}}, \tag{14}$$

where $i_f, i_r$ are the mounting angle. By integrating along the wing span, the total aerodynamic force $Z_a$ along the Z-axis can be obtained from the sum of the aerodynamic lift $L_f, L_r$ of the front and rear levitation wing.

$$Z_a = L_f + L_r \tag{15}$$

$$L_f = \frac{1}{2}\rho V_{x0}^2 c_f \int_{-b_f/2}^{b_f/2} C_{Lfi}(y_i)dy_i \tag{16}$$

$$L_r = \frac{1}{2}\rho V_{x0}^2 c_f \int_{-b_r/2}^{b_r/2} C_{Lri}(y_i)dy_i. \tag{17}$$

*2.3. Aerodynamic Force along the Y-Axis*

Given that the Aerotrain runs in a U-shaped guideway, the aerodynamic force generated from the guide wings are also under the WIG effect. The aerodynamic force $Y_a$ along $Y$-axis is the combination of the aerodynamic force from 4 guide wings $Y_{fl}, Y_{fr}, Y_{rl}, Y_{rr}$.

$$Y_a = Y_{fl} + Y_{fr} + Y_{rl} + Y_{rr}. \tag{18}$$

Using the same idea as the aerodynamic force in the Z-axis, where the distance $d_i$ plays the role of the height $h_i$ and $z_i$ plays the role of the spanwise $y_i$, the coefficient of differential force of the guide wing at the spanwise $z_i$, as shown in the Figures 8 and 9, could be written as follows [21]:

$$C_{yfli}(z_i) = a_1 \exp\left(-a_2 \frac{d_1(d,\theta,\phi,\psi,z_i)}{c_{sf}}\right) + a_3 \beta_{fl}(\beta,r) + a_4 \tag{19}$$

$$C_{yfri}(z_i) = -a_1 \exp\left(-a_2 \frac{d_2(d,\theta,\phi,\psi,z_i)}{c_{sf}}\right) - a_3 \beta_{fr}(\beta,r) - a_4 \tag{20}$$

$$C_{yrli}(z_i) = b_1 \exp\left(-b_2 \frac{d_3(d,\theta,\phi,\psi,z_i)}{c_{sr}}\right) + b_3 \beta_{rl}(\beta,r) + b_4 \tag{21}$$

$$C_{yrri}(z_i) = -b_1 \exp\left(-b_2 \frac{d_4(d,\theta,\phi,\psi,z_i)}{c_{sr}}\right) - b_3 \beta_{rr},(\beta,r) - b_4 \tag{22}$$

where $\psi$ is the yawing angle; $i_{fl}, i_{fr}, i_{rl}, i_{rr}$ are the mounting angle of the front left, front right, rear left, rear right guide wing relatively; $r$ is the yawing angular velocity; $d$ is the width of the guideway; $\beta$ is the sideslip angle; $\beta_{fl}, \beta_{fr}, \beta_{rl}, \beta_{rr}$ are the angle of attack of the guide wings.

$$\beta_{fl}(\beta,r) = \beta + i_{fl} - r\frac{l_f}{V_{x0}} \tag{23}$$

$$\beta_{fr}(\beta,r) = -\beta + i_{fr} + r\frac{l_f}{V_{x0}} \tag{24}$$

$$\beta_{rl}(\beta,r) = \beta + i_{rl} + r\frac{l_r}{V_{x0}} \tag{25}$$

$$\beta_{rr}(\beta,r) = -\beta + i_{rr} - r\frac{l_r}{V_{x0}}. \tag{26}$$

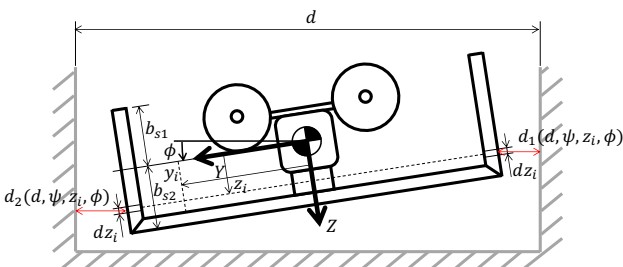

**Figure 8.** The dimension of the guide wing [21].

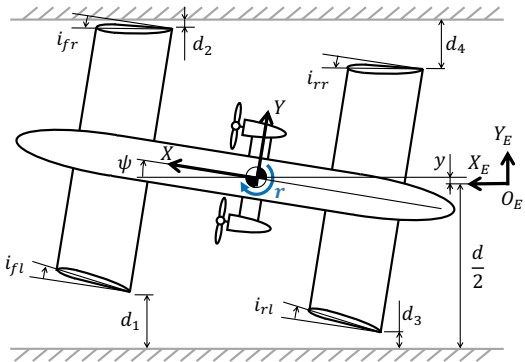

**Figure 9.** The top view of Aerotrain [21].

The distance $d_1, ..., d_4$ from the trailing edge of the guide wing to the wall could be calculated with respect to the inertial system $O_E - X_E Y_E Z_E$ as shown in Figure 10. The position vector of a point on the guide wing in the inertial system $^E p_i$ could be obtained from the position vector in the body-fixed system $^B p_i$ by the homogeneous translation matrix $T(d, \phi, \theta, \psi)$ as follows:

$$\begin{bmatrix} ^E p_i \\ 1 \end{bmatrix} = T(d, \phi, \theta, \psi) \begin{bmatrix} ^B p_i \\ 1 \end{bmatrix} \tag{27}$$

$$T(d, \phi, \theta, \psi) = \begin{bmatrix} ^E R_B(\phi, \theta, \psi) & ^E p_B(d) \\ 0 & 1, \end{bmatrix} \tag{28}$$

where $^E R_B$ is the rotation matrix and $^E p_B(d)$ is the position vector of the origin of the body-fixed system with respect to the inertial system.

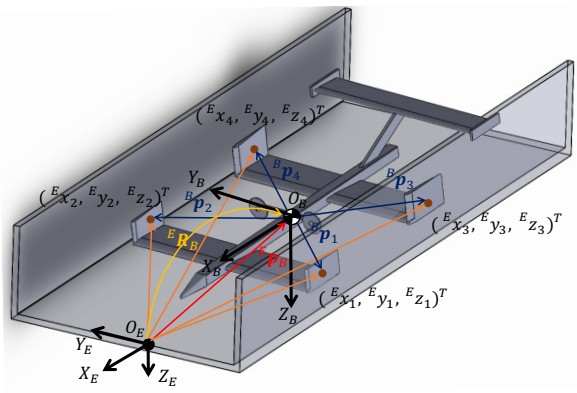

**Figure 10.** Position vector of the point on the guide wing with respect to the original of the inertial system [21].

In the body-fixed system, for example, the position vector of a point on the front-left guide wing at the span wise $z_i$ could be written as follows:

$$^B p_1(z_i) = \left[ l_f, -\frac{b}{2}, z_i \right]^T, \tag{29}$$

while in the inertial system, that position vector could be written as:

$$\begin{bmatrix} ^E p_1(d, \theta, \phi, \psi, z_i) \\ 1 \end{bmatrix} = [^E x_1(d, \theta, \phi, \psi, z_i), \quad ^E y_1(d, \theta, \phi, \psi, z_i), \quad ^E z_1(d, \theta, \phi, \psi, z_i), \quad 1]^T. \tag{30}$$

Using the Equations (27)–(30), the distance $d_1, ..., d_4$ could be calculated.

$$d_1(d, \theta, \phi, \psi, z_i) = \frac{d_{wall}}{2} + {}^E y_1(d, \theta, \phi, \psi, z_i) \tag{31}$$

$$d_2(d, \theta, \phi, \psi, z_i) = \frac{d_{wall}}{2} - {}^E y_2(d, \theta, \phi, \psi, z_i) \tag{32}$$

$$d_3(d, \theta, \phi, \psi, z_i) = \frac{d_{wall}}{2} + {}^E y_3(d, \theta, \phi, \psi, z_i) \tag{33}$$

$$d_4(d, \theta, \phi, \psi, z_i) = \frac{d_{wall}}{2} - {}^E y_4(d, \theta, \phi, \psi, z_i). \tag{34}$$

Therefore, by integrating along the wing span as shown in Figure 8, the aerodynamic force along the $Y$-axis of 4 guide wings could be expressed as follows:

$$Y_{fl} = \frac{1}{2}\rho V_{x0}^2 c_{sf} \int_{-b_{s1f}}^{b_{s2f}} C_{yfli}(z_i)dz_i \tag{35}$$

$$Y_{fr} = \frac{1}{2}\rho V_{x0}^2 c_{sf} \int_{-b_{s1f}}^{b_{s2f}} C_{yfri}(z_i)dz_i \tag{36}$$

$$Y_{rl} = \frac{1}{2}\rho V_{x0}^2 c_{sr} \int_{-b_{s1r}}^{b_{s2r}} C_{yrli}(z_i)dz_i \tag{37}$$

$$Y_{rr} = \frac{1}{2}\rho V_{x0}^2 c_{sr} \int_{-b_{s1r}}^{b_{s2r}} C_{yrri}(z_i)dz_i, \tag{38}$$

where $c_{sf}, c_{sr}$ are the chord length of the front and rear guide wings.

### 2.4. Aerodynamic Moment about the $X, Y, Z$ Axes

The aerodynamic moment about the $Y$-axis (pitching moment) $M_a$ is the sum of the moment of two levitation wings, the moment of the fuselage and the moment cause by the lift of these wings.

$$M_a = M_f + M_r + M_{fus} + l_f L_f - l_r L_r, \tag{39}$$

where $M_f, M_r$ are the moments of the front and rear levitation wings, and it can be obtained by integrating along the wingspan as can be seen in Figure 6.

$$M_f = \frac{1}{2}\rho V_{x0}^2 c_f \int_{-b_f/2}^{b_f/2} C_{Mfi}(y_i)y_idy_i \tag{40}$$

$$M_r = \frac{1}{2}\rho V_{x0}^2 c_r \int_{-b_r/2}^{b_r/2} C_{Mri}(y_i)y_idy_i. \tag{41}$$

The aerodynamic moment coefficient $C_{Mfi}(y_i), C_{Mri}(y_i)$ about the $Y$-axis at the spanwise $y_i$ of two levitation wings are also under the WIG effect, therefore it can be computed from the Equations (6)–(8) as follows [15]:

$$C_{Mfi}(y_i) = a_5 \exp\left(-a_6\frac{h_f}{c_f}\right) + a_7 \exp\left(-a_8\frac{h_f}{c_f}\right) + a_9\alpha_f + a_{10} \tag{42}$$

$$C_{Mri}(y_i) = b_5 \exp\left(-b_6\frac{h_r}{c_r}\right) + b_7 \exp\left(-b_8\frac{h_r}{c_r}\right) + b_9\alpha_r + b_{10}, \tag{43}$$

where $a_5, ..., a_{10}, b_5, ..., b_{10}$ are the obtained constants from the experimental results [20]. The constants are different between each levitation wing because of the generated downwash behind the front levitation wing.

The moment of the fuselage $M_{fus}$ is expressed as follows [22]:

$$M_{fus} = \rho V_{x0}^2 V_{fus}\alpha, \tag{44}$$

where $V_{fus}$ is the volume of the equivalent fuselage [22] of the Aerotrain.

The moment about the *X*-axis (rolling moment) $L_a$ is obtained from the aerodynamic lift from the main levitation wings $L_{lw}$, the front guide wings $L_{gwf}$, the rear guide wing $L_{gwr}$ [21], and the roll damping moment caused by yaw rate of the front levitation wing $L_{rf}$ and rear levitation wings $L_{rr}$.

$$L_a = L_{lw} + L_{gwf} + L_{gwr} + L_{rf} + L_{rr} \tag{45}$$

$$L_{lw} = -\frac{1}{2}\rho V_{x0}{}^2 \left( c_f \int_{-b_f/2}^{b_f/2} C_{Lfi}(y_i)y_i dy_i + c_r \int_{-b_r/2}^{b_r/2} C_{Lri}(y_i)y_i dy \right) \tag{46}$$

$$L_{gwf} = \frac{1}{2}\rho V_{x0}{}^2 c_{sf} \int_{-b_{s1f}}^{b_{s2f}} \left( -C_{yfli}(z_i) + C_{yfri}(z_i) \right) z_i dz_i \tag{47}$$

$$L_{gwr} = \frac{1}{2}\rho V_{x0}{}^2 c_{sr} \int_{-b_{s1r}}^{b_{s2r}} \left( -C_{yrli}(z_i) + C_{yrri}(z_i) \right) z_i dz_i. \tag{48}$$

In rolling motion, the aerodynamic lift of the main levitation wings has the derivation from roll rate $p = d\phi/dt$. As can be seen in the Figure 11, the positive roll rate induces a positive angle of attack at the right-wing tip of $\Delta\alpha_p(b/2) = pb/2V$. Because of roll rate is occurring about the *X*-axis, the induced angle of attack varies linearly from $\Delta\alpha_p = +pb/2V$ at the right wing tip to $-pb/2V$ at the left wing tip [23]. Therefore, the angle of attack of the front levitation wing and rear levitation wing in lift coefficient $C_{Lfi}, C_{Lri}$ could be written as:

$$\alpha_f(\alpha, q, p, i) = \alpha + i_f - q\frac{l_f}{V_{x0}} + p\frac{y_i}{2V_{x0}} \tag{49}$$

$$\alpha_r(\alpha, q, p, i) = \alpha + i_r + q\frac{l_r}{V_{x0}} + p\frac{y_i}{2V_{x0}}. \tag{50}$$

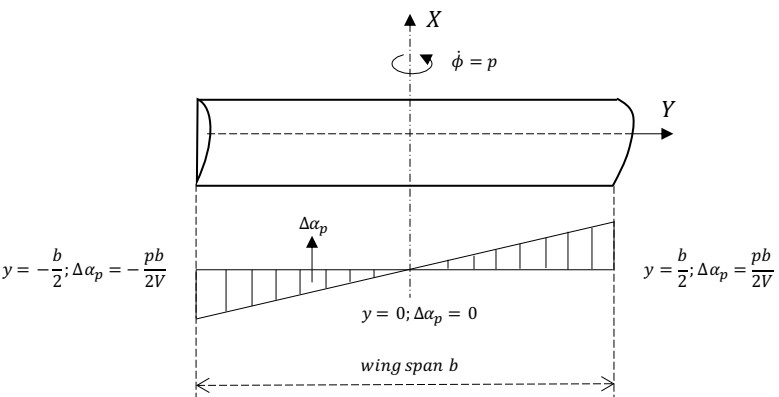

**Figure 11.** Airplane wing in rolling flight (adapted from Reference [23]).

The roll damping moment caused by yaw rate is defined by [23]:

$$L_r = \frac{QSb}{I_{xx}} \left( \frac{b}{2V} \right) C_{l_r}, \tag{51}$$

where $Q = \frac{1}{2}\rho V_{x0}{}^2$ is the dynamic pressure, $I_{xx}$ is the moment of inertial in *X*-axis, *V* is the airspeed, $C_{l_r}$ is the yaw rate derivative in the roll damping moment.

Without the vertical tail, as an initial approximation, based on elementary wing strip theory approach, the yaw rate derivative in the roll damping moment $C_{l_r}$ could be written as [23]:

$$C_{l_r} = \frac{C_L}{4}. \tag{52}$$

Therefore, the rolling damping moment caused by the yaw rate of the front and rear levitation wings could be written as:

$$L_{rf} = \frac{b_f^2}{8V_{x0}I_{xx}}L_f \tag{53}$$

$$L_{rr} = \frac{b_r^2}{8V_{x0}I_{xx}}L_r. \tag{54}$$

The aerodynamic moment about the *Z*-axis (yawing moment) $N_a$ is the total moment of the guide wings, the moment of the side force from the guide wings and the moment of the fuselage:

$$N_a = N_{fl} + N_{fr} + N_{rl} + N_{rr} + l_f(Y_{fl} + Y_{fr}) - l_r(Y_{rl} + Yrr) + N_{fus}, \tag{55}$$

where $N_{fl}, N_{fr}, N_{rl}, N_{rr}$ are the moments of guide wings under WIG effect and could be computed in the same way as the moment of the levitation wings by integrating along the wingspan; $N_{fus}$ is the moment of the fuselage.

$$N_{fl} = \frac{1}{2}\rho V_{x0}^2 c_{sf} \int_{-b_{s1f}}^{b_{s2f}} C_{Mfli}(z_i)z_i dz_i \tag{56}$$

$$N_{fr} = \frac{1}{2}\rho V_{x0}^2 c_{sf} \int_{-b_{s1f}}^{b_{s2f}} C_{Mfri}(z_i)z_i dz_i \tag{57}$$

$$N_{rl} = \frac{1}{2}\rho V_{x0}^2 c_{sr} \int_{-b_{s1r}}^{b_{s2r}} C_{Mrli}(z_i)z_i dz_i \tag{58}$$

$$N_{rr} = \frac{1}{2}\rho V_{x0}^2 c_{sr} \int_{-b_{s1r}}^{b_{s2r}} C_{Mrri}(z_i)z_i dz_i. \tag{59}$$

As was mentioned in the introduction, the guide wings is also under WIG effect. Therefore, the aerodynamic moment coefficients $C_{Mfli}, C_{Mfri}, C_{Mrli}(z_i), C_{Mrri}(z_i)$ of the guide wings at the wingspan $z_i$ can be computed from the Equations (6)–(8) as follows:

$$C_{Mfli}(z_i) = a_5 \exp\left(-a_6\frac{d_1}{c_{fl}}\right) + a_7 \exp\left(-a_8\frac{d_1}{c_{fl}}\right) + a_9\beta_{fl} + a_{10} \tag{60}$$

$$C_{Mfri}(z_i) = a_5 \exp\left(-a_6\frac{d_2}{c_{fr}}\right) + a_7 \exp\left(-a_8\frac{d_2}{c_{fr}}\right) + a_9\beta_{fr} + a_{10} \tag{61}$$

$$C_{Mrli}(z_i) = b_5 \exp\left(-b_6\frac{d_3}{c_{rl}}\right) + b_7 \exp\left(-b_8\frac{d_3}{c_{rl}}\right) + b_9\beta_{rl} + b_{10} \tag{62}$$

$$C_{Mrri}(z_i) = b_5 \exp\left(-b_6\frac{d_4}{c_{rr}}\right) + b_7 \exp\left(-b_8\frac{d_4}{c_{rr}}\right) + b_9\beta_{rr} + b_{10}, \tag{63}$$

where $a_5, ..., a_{10}, b_5, ..., b_{10}$ are obtained constants from the experimental results [20]. The guide wings at the front and rear levitation wings have different constants because of the downwash occurs when the airflow passes the front guide wings.

The moment of the fuselage $N_{fus}$ is expressed as follows [22]:

$$N_{fus} = \rho V_{x0}{}^2 V_{fus} \beta, \tag{64}$$

where $V_{fus}$ is the volume of the equivalent fuselage [22] of the Aerotrain.

### 2.5. Horizontal Tail Model

The lift coefficient and moment coefficient of the horizontal tail is a function of the angle of attack and the angular velocity [19]. Depending on the angle of attack $\alpha$, the lift coefficient $C_{Lht}$ is directly related to the mounting angle $i_{ht}$ and the downwash angle $\varepsilon$ caused by the air flowing through the main wing.

$$C_{Lht} = \alpha_{ht}(\alpha - \varepsilon - i_{ht}). \tag{65}$$

In the case of the Aerotrain, the horizontal tail is located at the rear side of the train and at a high location so that the downwash flow does not affect the tail. Thus, the downwash angle is zero, $\varepsilon = 0$. Another assumption is that the horizontal tail has symmetry airfoil and does not generate any aerodynamic force when the angle of attack is zero, which means that the chord line is parallel to the $X$-axis in the body-fixed system or the mounting angle $i_{ht} = 0$. As for the thin, symmetric airfoil [24], the lift coefficient could be written as follows:

$$C_{Lht}(\alpha) = 2\pi\alpha. \tag{66}$$

The horizontal tail is located at a distance $l_{ht}$ from the origin of the body-fixed coordinate system to the aerodynamic center of the tail as shown in Figure 12. Thus, the effect of pitching rate $q$ in terms of the changing of angle of attack $\alpha$ is described as:

$$\Delta\alpha_{ht} = q\frac{l_{ht}}{V_{x0}}. \tag{67}$$

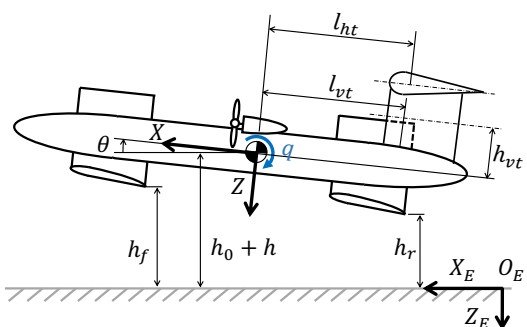

**Figure 12.** Horizontal and vertical tail.

Thus, the lift coefficient of the horizontal tail caused by the pitching rate is:

$$C_{Lht}(q) = 2\pi q\frac{l_{ht}}{V_{x0}}. \tag{68}$$

Therefore, the total lift force coefficient is the sum of these two coefficients:

$$C_{Lht}(\alpha, q) = C_{Lht}(\alpha) + C_{Lht}(q). \tag{69}$$

The total pitching moment from the horizontal tail is also a function of the angle of attack $\alpha$ and the pitching rate $q$ and it could be written as:

$$C_{Mht}(\alpha, q) = -l_{ht}C_{Lht}(\alpha) - l_{ht}C_{Lht}(q). \tag{70}$$

*2.6. Vertical Tail Model*

Similar to the horizontal tail model, we assume that the vertical tail has a symmetry airfoil, locates at a distance $l_{vt}$ from the CoG of the Aerotrain to the aerodynamic center of the tail as shown in Figure 12, and the chord line is align to the *X*-axis in the body-fixed system. Thus, the side force coefficient of the vertical tail $C_{Yvt}$ caused by the angle of attack $\beta$ and the yaw rate $r$ is described as:

$$C_{Yvt}(\beta, r) = C_{Yvt}(\beta) + C_{Yht}(r) \tag{71}$$

$$C_{Yvt}(\beta) = 2\pi\beta \tag{72}$$

$$C_{Yht}(r) = 2\pi r \frac{l_{vt}}{V_{x0}}. \tag{73}$$

The side force of the vertical tail also contributes a rolling moment:

$$L_{vt} = \frac{1}{2}h_{vt}\rho S_{vt}V_{x0}{}^2 C_{Yvt}, \tag{74}$$

where $h_{vt}$ is the distance from the *X*-axis of the body-fixed system to the 1/2 vertical tail wingspan as shown in Figure 12, $S_{vt}$ is the vertical tail area.

The total yawing moment from the vertical tail is also the function of the angle of attach $\beta$ and the yawing rate $r$ and it could be written as:

$$C_{Nht}(\beta, r) = -l_{vt}C_{Yvt}(\beta) - l_{vt}C_{Yvt}(r). \tag{75}$$

*2.7. Dihedral Levitation Wing Model*

The dihedral wing design is easily found in the commercial and training aircraft because of its efficient rolling damping, that is important to keep the aircraft stable. When a rolling motion happens, it also generate a coupled yawing motion, and results a side slip angle $\beta$. Assumed that the dihedral angle $\Gamma$ is small enough, the differential angle of attack $\Delta\alpha$ [23] on the left and right panel of the levitation wing change in opposite way.

$$\Delta\alpha = \Gamma\beta. \tag{76}$$

Thus, the difference amount of lift on the left and the right panel of the levitation wing generates a recovering moment. In dihedral design of the front wing, the angle of attack of the front left levitation wing $\alpha_{fl}$ and front right levitation wing $\alpha_{fr}$ in the Equation (13) could be written as:

$$\alpha_{fl}(\alpha, q) = \alpha + i_f - q\frac{l_f}{V_{x0}} - \beta\Gamma \tag{77}$$

$$\alpha_{fr}(\alpha, q) = \alpha + i_r + q\frac{l_r}{V_{x0}} + \beta\Gamma. \tag{78}$$

Figure 13 shows the height of the front left wing $h_{fl}$ and front right wing $h_{fr}$ in dihedral wing design. The height of the front wing in the Equation (11) could be written as:

$$h_{fl}(h, \theta, \phi, y_i) = h_0 + h + l_f\theta + y_i\frac{\tan\Gamma + \phi}{1 - \phi\tan\Gamma} \tag{79}$$

$$h_{fr}(h, \theta, \phi, y_i) = h_0 + h + l_f\theta + y_i\frac{\tan\Gamma - \phi}{1 + \phi\tan\Gamma}. \tag{80}$$

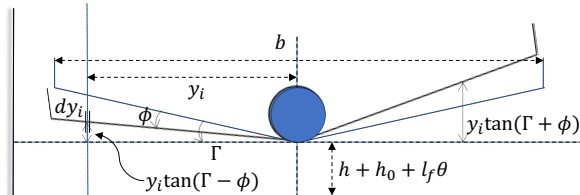

**Figure 13.** Dihedral levitation wing.

## 3. Simulation and Discussion

To evaluate the dynamic stability of the Aerotrain, a three-dimensional dynamic model has been developed in the MapleSim [25] environment using a rigid body model as shown in Figure 14. The advantage of the MapleSim over the others is its friendly user interface and its sufficiency when creating custom components that are not available in the default library. Moreover, by improving the detail of each component (for example, the roll rate and yaw rate derivatives in the rolling moment), the accuracy of the previous model proposed in Reference [21] could be enhanced with unnoticed additional computational cost.

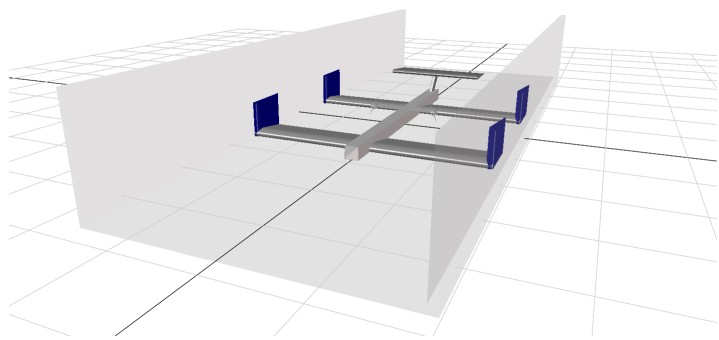

**Figure 14.** Aerotrain in MapleSim.

Base on the mathematical equations, the aerodynamic components, such as lift and moment of the levitation wings and the guide wings has been created as the customer components. The total net force in the $Z$-axis which includes the lift from the levitation wing, the horizontal tail and the total net force in $Y$-axis includes the side force from the guide wings and the vertical tail were applied to the Aerotrain body through the "Apply World Force" component. Similar to the force, the total net moment about $X, Y, Z$ axes was also applied to the body of the train through the "Apply World Moment" component.

As a case study, the parameters used in the simulation in Table 1 were approximate from the experimental ARTE02 prototype developed in Reference [16]. The mass of the prototype was measured while the other parameters are designed values. The prototype, as shown in Figures 15–17, is equipped with four aileron flaps which control the pitching and rolling motion, and four rudders which control the yawing motion according to the distance between the vehicle body and the guideway which is measured from the laser displacement sensors. By using physical parameters of ARTE02 prototype, the evaluation of the state space equation of the dynamic model of Aerotrain was shown that the system is unstable but controllable [16].

At the equilibrium state, according to each design of the Aerotrain, the velocity $V_{x0}$ is computed, so that the generated aerodynamic lift is equal to the vehicle's weight; by moving the CoG of Aerotrain along $X$-axis of the body-fixed coordinate system, the distance $l_f$ of the front levitation wing and $l_r$ of the rear levitation wing are calculated, so that the total moment applies to the model is zero. In order to evaluate the dynamic stability of the Aerotrain, during 25 s of the simulation time, the external disturbances were applied to the body after 2 s to observe the responsibility of the model. The external

disturbances include a force of 50 *N* which applied in *Z* axis, two moments of 10 *N* · *m* which applied to rotate the body about the *X*-axis and *Z*-axis.

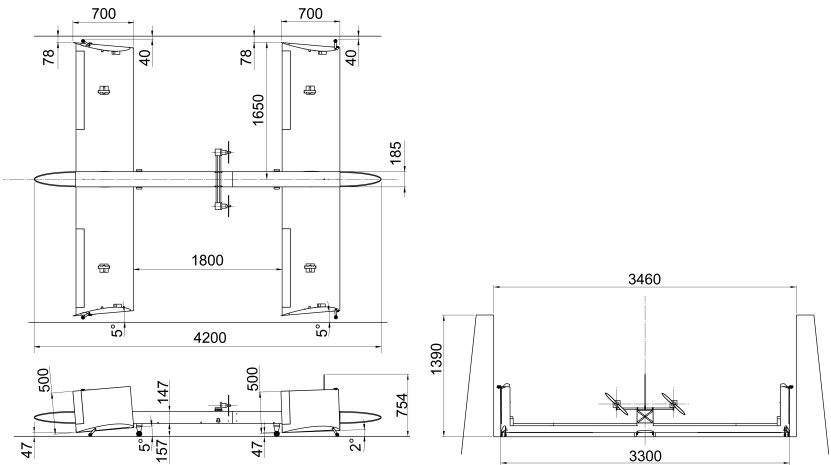

**Figure 15.** ARTE02 dimensions [16].

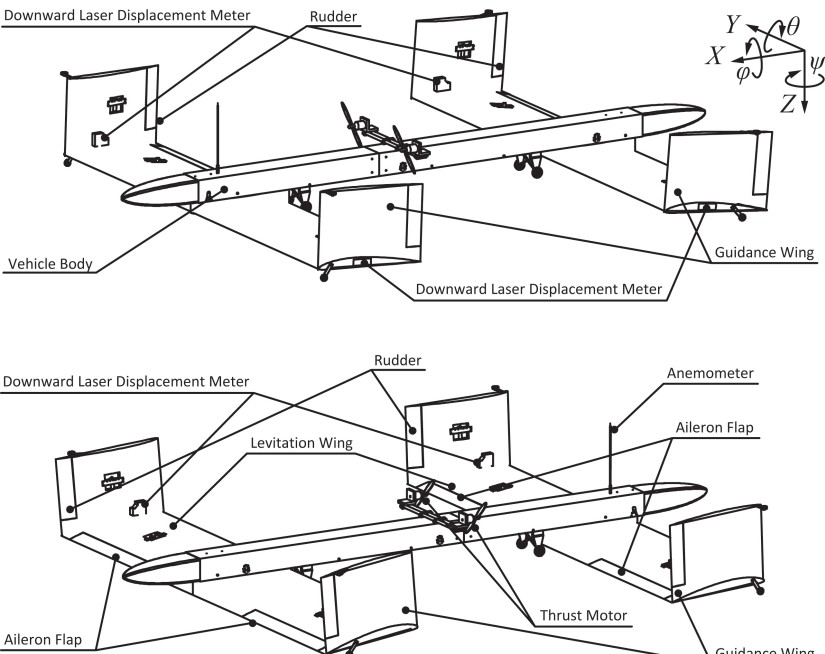

**Figure 16.** ARTE02 isometric view [16].

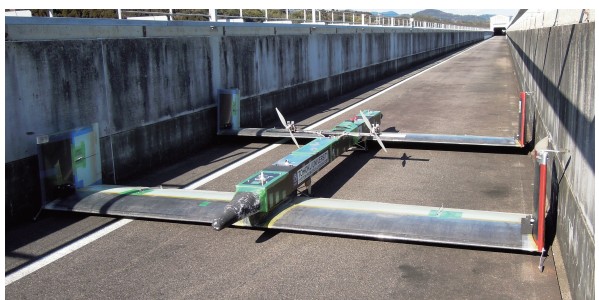

**Figure 17.** The experimental WIG effect vehicle ARTE02 [16].

**Table 1.** Aerotrain parameters.

| Parameters | Value |
|---|---|
| Mass [kg] | 42.5 |
| Wing area [m$^2$] | 2.31 |
| Chord length [m] | 0.7 |
| Wing span [m] | 3.3 |
| Levitation height [m] | 0.07 |
| $i_f$ [deg] | 1.48 |
| $i_r$ [deg] | 2.63 |
| Horizontal tail area $S_{ht}$ [m$^2$] | 0.5 |
| Horizontal tail location $l_{ht}$ [m] | 2.5 |
| Vertical tail area $S_{vt}$ [m$^2$] | 0.5 |
| Vertical tail location $l_{vt}$ [m] | 2.5 |
| Z impact [$N$] | 50 |
| Roll impact [$N \cdot m$] | 10 |
| Yaw impact [$N \cdot m$] | 10 |

### 3.1. Longitudinal Dynamic Stability

With the design of the experimental prototype ARTE02, at the equilibrium state, it requires a velocity of $V_{x0} = 34.65$ m/s, and the CoG of the vehicle is located at a location such that $l_f = 0.84$ m, $l_r = 1.69$ m. A horizontal tail with the area of $S_{ht} = 0.5$ m$^2$, located at $l_{ht} = 2.5$ m was added to the model. At the same condition, with the horizontal tail, the displacement in the Z-axis and pitch angle $\theta$ have been convergent as shown in Figures 18 and 19 thanks to the spring element $\alpha$ and damper element $q$. The simulation results have clearly proved that the effectiveness of the horizontal tail still remained in the three-dimensional model.

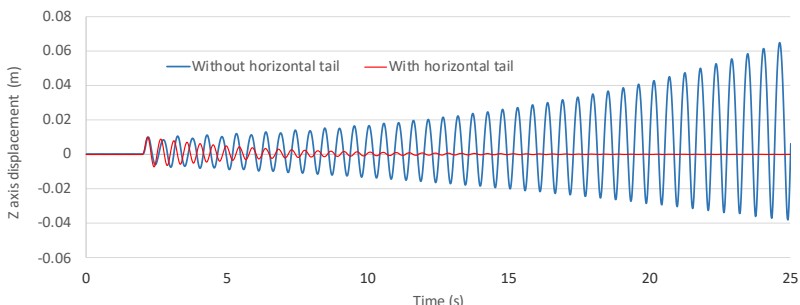

**Figure 18.** Displacement in Z-axis.

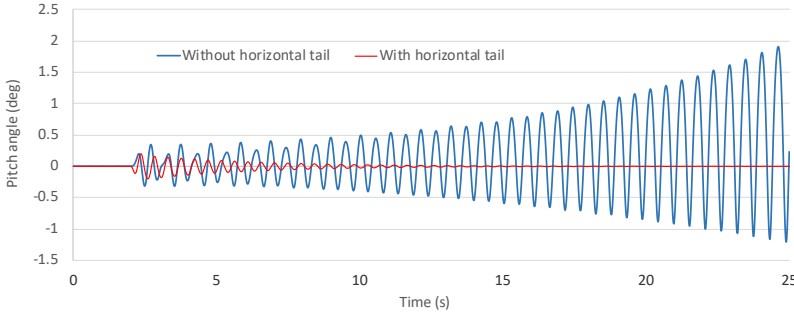

**Figure 19.** Pitch angle.

In Wang's study [26], it was found that the configuration with a higher horizontal tail has insignificant effects on the aerodynamic efficiency of a WIG craft. Thus, based on the above

confirmation, the design of the horizontal tail (area $S_{ht}$ and location $l_{ht}$) is targeted to investigate the relationship between the design and the longitudinal dynamic stability of the Aerotrain. First, the horizontal tail area is fixed as 0.5 m², the distance from the CoG of the Aerotrain to the aerodynamic center of the horizontal tail $l_{ht}$ ranges from 1 m to 5 m at a 0.25 m step. As can be seen in Figure 20, the response of the Aerotrain in $Z$-axis is different at each location. Especially, at the location of 1 m the displacement in the $Z$-axis becomes divergent, which means the design does not have longitudinal dynamic stability.

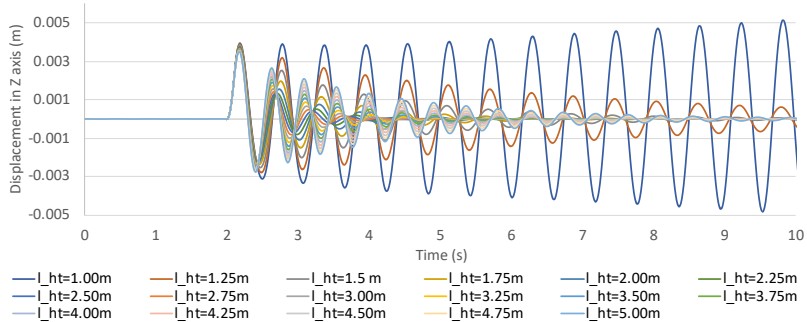

**Figure 20.** $Z$-axis displacement at different horizontal tail locations with the tail area of $S_{ht}$ = 0.5 m².

Zooming in the detail of the Figure 20 at the range from 2 s to 3 s, Figure 21 shows that after one period, the location of 2.25 m and 2.5 m have the highest amplitude reduction, therefore have the best performance in terms of dynamic stability. In order to investigate the relationship between the location $l_{ht}$, the area $S_{ht}$ and the longitudinal dynamic stability, $T_{1/2}$ has been introduced. $T_{1/2}$ is the time to damp to half amplitude from maximum amplitude, so that smaller $T_{1/2}$, better dynamic stability.

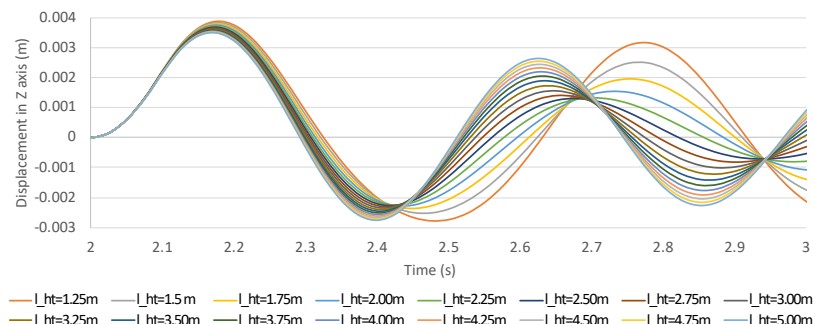

**Figure 21.** $Z$-axis displacement at different horizontal tail locations with the tail area of $S_{ht} = 0.5$ m² from 2 s to 3 s.

At the same range of the horizontal tail location, two more different horizontal tail area of 1 m² and 1.5 m² have been simulated. The relationship between the horizontal tail location, area and the dynamic stability represented by $T_{1/2}$ has been clarified in Figures 22 and 23. As a simple prediction, the ideal design of the horizontal tail could be described in a straight line form:

$$S_{ht} = -0.3778l_{ht} + 1.8654. \tag{81}$$

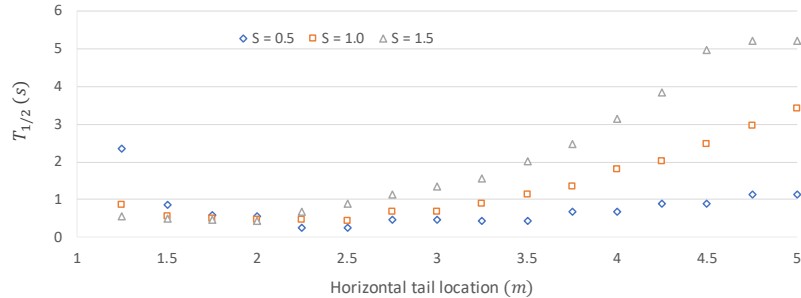

**Figure 22.** Relationship between $T_{1/2}$ and the horizontal tail location with 3 tail area values.

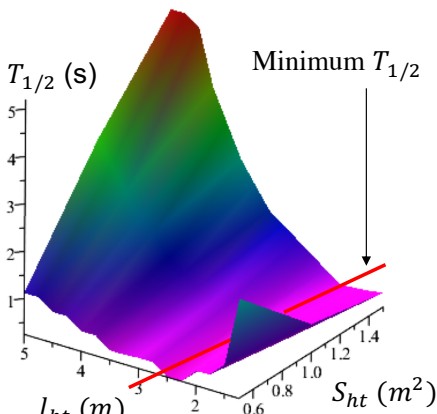

**Figure 23.** Relationship between $T_{1/2}$, the horizontal tail location and area.

### 3.2. Lateral Dynamic Stability

The response of the Aerotrain to the rolling disturbance of 10 $N \cdot m$ is showed in the Figure 24. The simulation result shows that the roll angle $\phi$ has been convergent, or the design of flat levitation wing of Aerotrain can maintain lateral dynamic stability. The result again confirms the advantage of WIG effect in lateral stability of WIG effect vehicles, that was found through the numerical studies by Amir [27]. However, the damping is insufficient to damp the vibration in a short time. As a simple comparison, the front levitation wing has been replaced by a dihedral wing with $\Gamma = 10$ degrees while maintaining the same wingspan in order to fit into the guideway. The dihedral wing design caused a reduction of the aerodynamic lift at the front levitation wing. Thus, it requires a velocity of 42.8 km/h (increased 23.5% in comparison with the flat levitation wing design) to compensate that reduction. Also, the reduction of the aerodynamic lift at the front levitation wing shifted the CoG of the vehicle rearward, so that $l_f = 1.31$ m, $l_r = 1.22$ m to maintain the equilibrium state.

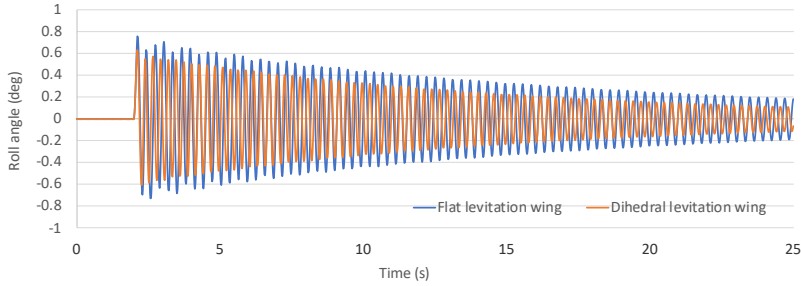

**Figure 24.** Roll angle.

In the dihedral levitation wing design, the vibration amplitude is smaller than that of the flat levitation wing design, but the trend is the same. Simulation result shows that the dihedral levitation wing design can increase the stability as it does on the airplane. However, to trade off with that small increasing of the stability, with the same wing span, the dihedral levitation wing requires a higher velocity to compensate the reduction of the aerodynamic lift at the front levitation wing. Therefore, it can be said that the dihedral levitation wing design is not suitable for the Aerotrain application. The lateral stability of the Aerotrain still needs the assist of the control system, especially for the damping.

### 3.3. Directional Dynamic Stability

By keeping a horizontal tail with the area of $S_{ht} = 0.5$ m$^2$, located at $l_{ht} = 2.5$ m, a vertical tail with the area of $S_{vt} = 0.5$ m$^2$, located at $l_{vt} = 2.5$ m, was added to the model. As can be seen in Figures 25 and 26, the design without the vertical tail can achieve the directional stability. However, the directional damping is insufficient. At the same condition, with the vertical tail, the displacement in the $Y$-axis and yaw angle has significantly reduction of the vibration amplitude.

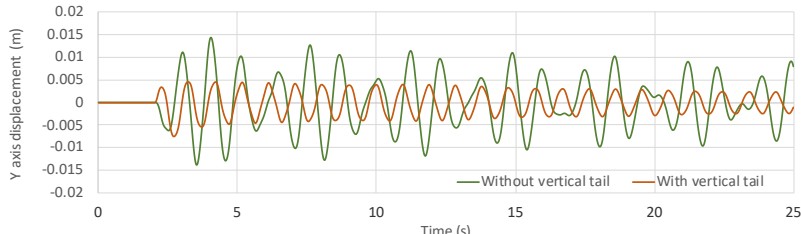

**Figure 25.** Displacement in $Y$-axis.

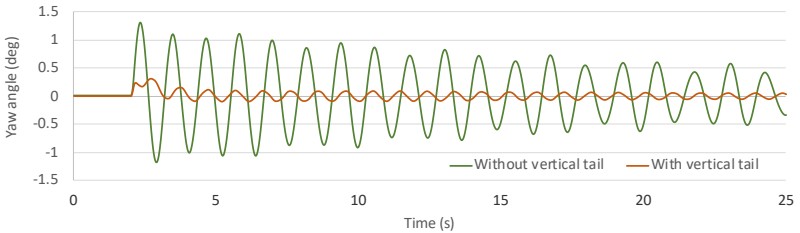

**Figure 26.** Yaw angle.

In order to investigate the vertical tail design (vertical tail location $l_{vt}$ and area $S_{vt}$) and the directional dynamic stability, various designs has been simulated. First, at a fixed area of $S_{vt} = 0.5$ m$^2$, the location of the vertical tail ranges from 1.5 m to 5 m at 0.25 m step. The simulation results have been shown in Figure 27. Since the displacement in the $Y$-axis and the yaw angle show the same trend of the vibration amplitude reduction, we can use one of them to evaluate the effect of the vertical tail. In this paper, the yaw angle was used. At different location, the maximum amplitude of the Yaw angle change accordingly. Therefore, the maximum amplitude $A_{Max}$ is used to evaluate the design of the vertical tail instead of $T_{1/2}$, that was used in case of the horizontal tail. It could be said that the smaller $A_{Max}$, the better directional stability.

At the same range of the vertical tail location, two more different vertical tail area of $S_{vt} = 1.0$ m$^2$ and $S_{vt} = 1.5$ m$^2$ have been simulated. The relationship between the vertical tail location $l_{vt}$, area $S_{vt}$ and the directional dynamic stability represented by $A_{Max}$ has been clarified in Figures 28 and 29. The larger vertical tail area, the better directional stability and (or) the farther vertical tail location, the better stability.

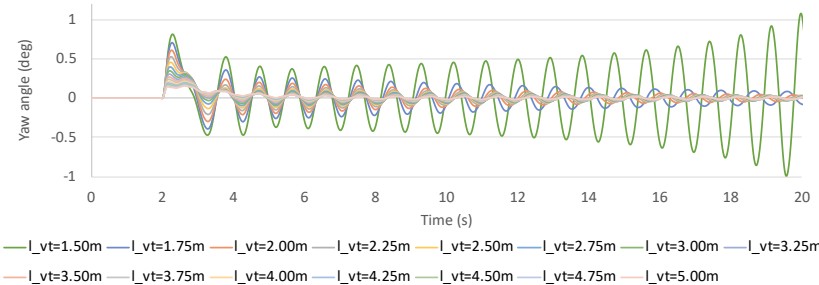

**Figure 27.** Yaw angle at different vertical tail locations with $S_{vt} = 0.5$ m$^2$.

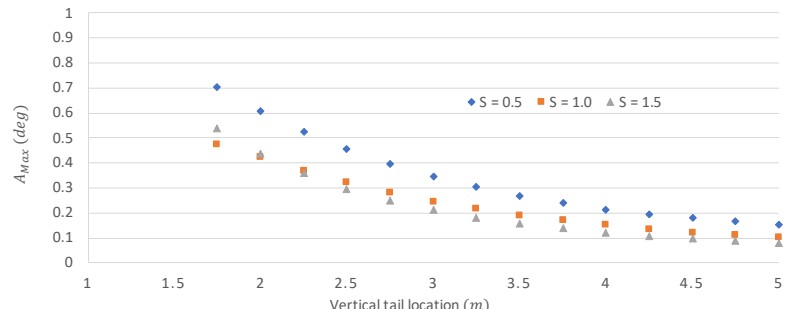

**Figure 28.** Relationship between $A_{Max}$ and the vertical tail location with 3 tail area values.

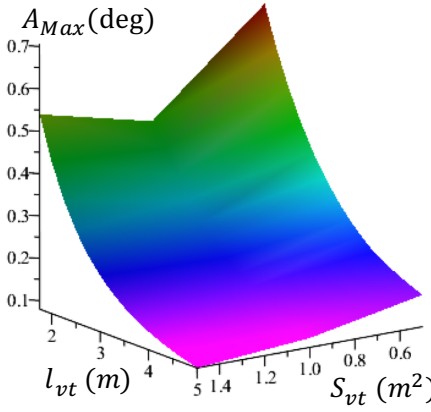

**Figure 29.** Relationship between $A_{Max}$, vertical tail location and vertical tail area.

## 4. Conclusions and Future Work

In this paper, the three-dimensional dynamic model of the Aerotrain has been developed taking into account the nonlinear aerodynamic forces and moments under the WIG effect. Compared with the previous studies [19,21] the side force from the 4 guide wings, the aerodynamic lift and moment from the levitation wings, the horizontal tail and the vertical tail have been integrated into the rigid body model to make the dynamic model more specific so that it will be useful for the optimization studies.

Based on this dynamic model, the design-stability relation of the Aerotrain has been investigated. The horizontal tail with several designs in tail location and tail area has been simulated to evaluate the longitudinal dynamic stability. The dihedral levitation wing design has been considered to evaluate the lateral dynamic stability. As for the directional dynamic stability, several designs of the vertical tail in tail location and tail area has been considered as well.

The area and location of the horizontal tail have a great effect on the longitudinal stability. The horizontal tail shows its best performance if the tail area is a function of the tail location.

The dihedral levitation wing, expected to improve the longitudinal stability, did not show significant change. The vertical tail showed its effectiveness in terms of directional stability as the maximum vibration was significantly reduced. The position and location of the vertical tail also have a great effect on the directional stability. Unlike the horizontal tail, the larger vertical tail area, the better directional stability and (or) the farther vertical tail location, the better stability. Therefore, the design of the vertical tail is depended on the application purpose.

In the future, based on this dynamic model, the force in X-axis will be considered to complete the 6-degrees of freedom of the three-dimensional model, and the contribution of each effect on the model accuracy will be taken into account. The sensitivity analysis will be carried out by changing one parameter at a time and keeping all other parameters fixed to figure out what would be the consequences of parameters estimation bias to the model simulation accuracy. Figuring out which parameter is more sensitive than the others about the stability will benefit not only the initial design phase of Aerotrain but also the safety of the vehicle. Then, the next studies will be focused on the experiment of a small-scale prototype in order to evaluate the reliability of the dynamic model. The experimental results will be useful to develop a full-scale prototype and the dynamic model of a multi-cars design as well.

**Author Contributions:** Conceptualization, Q.H.L., J.J. and Y.S.; methodology, Q.H.L., J.J. and Y.S.; software, Q.H.L. and J.J.; writing—original draft preparation, Q.H.L.; writing—review and editing, Q.H.L., Y.S., D.M. and Y.T.; supervision, Y.S. and Y.T. All authors have read and agreed to the published version of the manuscript.

**Funding:** This work was partially supported by JSPS KAKENHI 26420814.

**Conflicts of Interest:** The authors declare no conflict of interest.

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
