# Peer review of "A Study on the Relationship between the Design of Aerotrain and Its Stability Based on a Three-Dimensional Dynamic Model â€"

_robotics, doi:10.3390/robotics9040096_

Round 1
Reviewer 1 Report
The paper presents a study on the relationship between the design of aerotrain and its stability based on a 3D dynamic model.
The content of the manuscript is of interest to the reader and is sound from a scientific point of view. The paper is well organized and easy to read.
Just a minor remark: all the figures have a "namefile.pdf" next to them, which should be deleted. Moreover, Figures 8 and 9 overlap.
Thus, I recommend acceptance of the paper, provided the minor changes mentioned above are taken into account.
Reviewer 2 Report
This paper addresses the dynamic modelling of an Aerotrain by taking into account nonlinear aerodynamic forces and moments under the Wing-In-Ground effect.
The proposed topic is interesting and suitable for this journal.
The paper content reports a quite detailed and well described model, which is an extension of previous works by the authors on this topic. Main novel contribution is give by including in the model several additional effects such as the aerodynamic lift and moment, which are generated by the levitation wings, and the effects of the horizontal vertical tails. Also authors reports interesting results of several simulations.
This reviewer finds the content valuable and mostly suitable for publication. If possible, authors should consider or at least discuss the following improvments:
It would be useful to compare more in details the previous models with the one proposed in this paper. This would allow to clearly identify the contribution of each effect on the proposed model accuracy and also from computational costs viewpoint (i.e. maybe if some effects can be neglected if they are adding little improvement to the accuracy while requiring significant additional computational costs).
The proposed simulations are relying on several parameters, which are listed in Table 1. It would be useful to discuss more in details how they have been identified. Moreover, a sensitivity analysis could be carried out to check what would be the consequences of parameters estimation errors/bias to the model simulation accuracy.
Also at least a couple of more sentences should be added on the controllability of the model, which here is not proven from theoretical viepoint.
reference list is quite relevant and suitable. However, self citation percentage is a bit high. Pls consider either to increase number of references from other sources or to reduce the number of self cited references.
In the conclusions authors mention the plan to make experimental tests in future. Would it be possible to make them with a scaled or with a full size prototype?
